# Estimating sales transitions between competing products via optimal transport

**Shoki Yamao**[ID]*, **Ryota Ueda, Shoichiro Koguchi, Michi Nakase, Aru Suzuki, Kohdai Toyoda, Ken Kobayashi**[ID], **Kazuhide Nakata**

Department of Industrial Engineering and Economics, School of Engineering, Institute of Science Tokyo, Meguro, Tokyo, Japan

* yamao.s.aa@m.titech.ac.jp

**Data availability statement:** The data used in this study was provided by Nikkei Inc. through the 2024 Data Analysis Competition, organized

## Abstract

In mature markets, where products are widely adopted, understanding how customers switch between competing products is crucial for companies to conduct effective marketing actions. However, due to privacy regulations, it is increasingly difficult to obtain point-of-sale (POS) data with individual customer identifiers (IDs). In this paper, we propose a method that estimates how sales shift between products using aggregated POS data without customer IDs. We formulate this as an optimal transport problem aimed at minimizing the total cost of brand-switching and introduce two regularization terms based on assumptions about sales transitions. We then solve the optimization problem with these regularizations using a projected gradient method.

We validated our approach on proprietary POS data from the Japanese beverage industry and found that the estimated transitions aligned with real market changes. For instance, during a liquor tax reform period, customers switched from products whose tax rates increased to those with lower rates. In the coffee market, many customers moved toward a newly launched brand. Although these results suggest that our method can capture market dynamics, the proprietary data limits reproducibility. In addition, the absence of customer IDs makes it impossible to track individual customer transitions. Incorporating such identifiers in future research could offer more deeper insights into consumer behavior.

## Introduction

### Background

In a mature market, where products are widely adopted, companies need to capture and maintain market share from competing products to survive [1]. In Japan's alcoholic beverage industry, for example, sales trends have remained flat over the past decade [2], and companies are under pressure to compete with existing customers to improve their performance. Similar dynamics can be seen in other industries, such as mobile telecommunications, newspapers, and automobiles, where companies are trying to acquire customers from competitors to maintain or increase their market share.

by the Joint Association Study Group of Management Science (https://jasmac-j.jimdofree.com/). Due to legal and contractual restrictions imposed by the Joint Association Study Group of Management Science, we are unable to publicly share the dataset. For any inquiries regarding data access, please contact the study group at jasmac.dac@gmail.com.

**Funding:** The author(s) received no specific funding for this work.

**Competing interests:** The authors have declared that no competing interests exist.

Given this situation, it is becoming more important for companies to understand how customers switch between brands and how products compete in the market. By tracking customer movement between products, companies can see their strengths and weaknesses and plan ways to attract more customers. Also, watching sales shifts within their own products can help spot unwanted competition between them. Looking at market-wide trends can show changes in customer preferences and competition.

### Related work

**Analysis of brand-switching behavior with customer IDs.** While existing studies on analyzing brand-switching behavior have long existed, they often rely on data tied to individual customers, typically obtained through surveys or customer ID tracking. For example, Tariq et al. [3] studied how advertising affects brand-switching behavior in Pakistan's soft drink market by surveying high school students. Similarly, Grover et al. [4] analyzed customer segments in the coffee market using an iterative Bayesian procedure with the dataset of 1,000 customers in Pittsfield, Massachusetts. Lam et al. [5] examined the impact of brand awareness on brand-switching behavior from 679 Spanish customers during the launch of the iPhone. Liao et al. [6] used the push-pull-mooring framework to examine factors promoting 500 Chinese smartphone users to switch brands. Al-Mashraie et al. [7] analyzed customer churn using multiple support vector machine classifiers on a dataset of 1,000,000 customers from a U.S. telecommunications company. Although such approaches can offer insights into customers' brand-switching behavior, they often require data that associates transactions with individual customer IDs, which is becoming increasingly difficult to obtain due to privacy regulations.

**Analysis of brand-switching behavior without customer IDs.** Recently, Chiba et al. [8] proposed a method for estimating purchase transitions from aggregated POS data, assuming product market share in a single period follows a steady-state distribution of a Markov chain. This method then estimates the transition probability matrix for each period on the basis of the observed market shares over multiple periods. They formulated the estimation problem as a linear optimization problem that minimizes the discrepancy between the transition probability matrices for consecutive periods under the constraint that the market share in each period follows a steady-state distribution. However, even if the observed market share changes over time, the proposed model includes a trivial optimal solution, where the transition matrices for all periods are identity matrices, which indicates no transitions occur between products within each period. Thus, this method can yield unrealistic outcomes and may fail to capture actual market dynamics.

### Our contribution

In this paper, we propose a new method for estimating sales transitions between products from POS data that does not include customer IDs by framing brand-switching as an optimal transport problem. Specifically, rather than considering estimated transaction matrices for each period as in Chiba et al. [8], we model the flow of sales from one period to the next, which enforces that changes in product shares must be reconciled through some nonzero transition. This naturally prevents the trivial solution, in which no transitions occur between products, from being an optimal solution unless consecutive market sales are exactly the same. We show the overview of our method in Fig 1.

**Optimal transport and proposed framework.** Optimal transport aims to find a plan to convert one probability distribution to another while minimizing the required transportation cost, and it has been successfully applied in various domains such as image processing

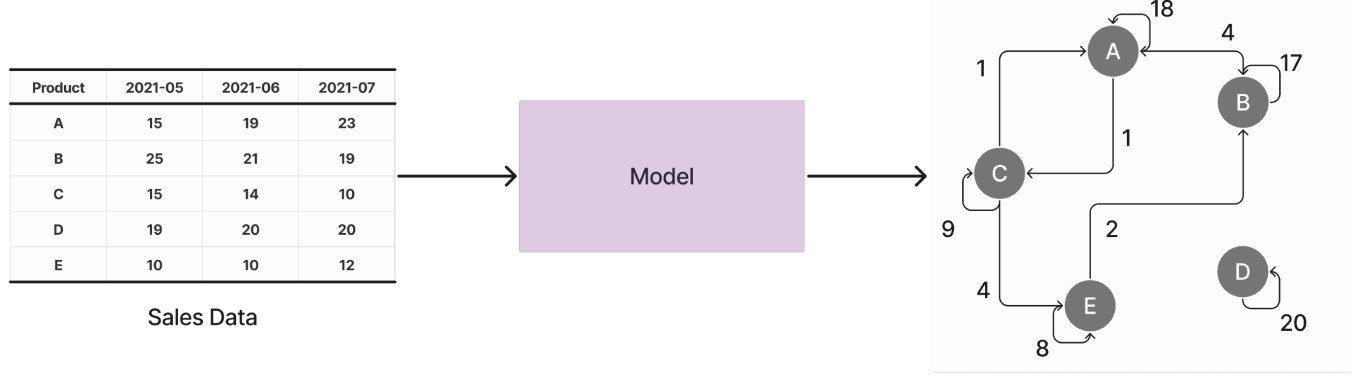

**Fig 1. Overview of the proposed method.** Given multiple periods of POS data, we estimate the sales transitions between products.

(e.g., color transfer [9]), natural language processing (e.g., loss optimization [10]), computer graphics (e.g., mass transport [11]), and economics (e.g., multi-agent matching [12]). Motivated by the success of optimal transport, we adopt this framework by interpreting each distribution as the allocation of sales across products and defining a cost structure that represents the difficulty of customers switching between products. Additionally, we introduce two regularization terms reflecting realistic market conditions: one assumes that the sales transition tendencies between the consecutive periods are similar, and the other assumes that customers' brand-switching behavior is not solely determined by cost and may involve some degree of stochastic uncertainty. We solve this regularized transport formulation using a projected gradient method. The main contributions of this study are summarized as follows:

- We formulate the problem of estimating sales transitions among products from aggregated POS data as an optimal transport problem, capturing transitions from one period to the next without requiring customer IDs.
- We introduce two regularization terms to ensure temporal consistency in sales transitions and account for stochastic uncertainty in purchasing behavior.
- We propose a projected gradient-based algorithm that efficiently solves the regularized optimal transport problem.
- We validate our approach using real POS data from Japan's beverage industry, demonstrating its ability to capture market dynamics, including tax effects and new product entries.

## Methods

In this section, we explain the proposed model for estimating the inflow and outflow of sales between products over time using multivariate time-series data of product sales. First, we formulate this problem as an optimal transport problem that minimizes the total cost of sales transitions between products. Next, we introduce two types of regularizations based on natural assumptions regarding purchasing behavior. Finally, we explain the solution algorithm based on the projected gradient method to solve the optimization problem with the two proposed regularization terms.

## Optimal transport problem formulation

Let a positive integer $T$ be the number of total periods and a positive integer $N$ be the number of products. The sales vector of all products in the period $t \in [T] := \{1, 2, \ldots, T\}$ is expressed by $s_t \in \mathbb{R}^N$. Given the set of sales vectors $\{s_t\}_{t=1}^T$, we estimate the sales transitions between products from the period $t$ to $t + 1$. Let $P_t \in \mathbb{R}^{N \times N}$ (for $t \in [T-1]$) represent the matrix that denotes the sales transitions between products from the period $t$ to period $t + 1$. The $(n,m)$ element of matrix $P_t$, denoted as $(P_t)_{nm}$, represents the total sales transition from the $n$-th product to the $m$-th product from the period $t$ to $t + 1$.

We consider the constraints for the sales transition matrix. Since each element of $P_t$ represents a sales transition, it must be non-negative:

$$P_t \geq O \quad (t \in [T-1]), \tag{1}$$

where $O$ is the zero matrix in $\mathbb{R}^{N \times N}$. The total sales outflow from the $n$-th product during the period from $t$ to $t + 1$ is assumed to be equal to the sales of the $n$-th product $n$ in the period $t$. Similarly, the total sales inflow to each product during the period from $t$ to $t + 1$ is assumed to be equal to the sales of the product in the period $t + 1$. These constraints are expressed as

$$P_t \mathbf{1} = s_t \qquad (t \in [T-1]), \tag{2}$$
$$\mathbf{1}^\top P_t = s_{t+1}^\top \qquad (t \in [T-1]), \tag{3}$$

where $\mathbf{1}$ is a vector with all elements equal to 1, and $\top$ denotes the transpose of a vector or matrix.

We then introduce the required cost for sales transitions between products from the period $t$ to $t + 1$. We set the cost required for sales transitions between products from the period $t$ to $t + 1$ as a constant matrix $C_t \in \mathbb{R}^{N \times N}$ ($t \in [T - 1]$) where the $(n,m)$ element of $C_t$, denoted as $(C_t)_{nm}$, represents the cost per unit sales incurred when switching from the $n$-th product to $m$-th product. Then, the total cost of sales transitions $P_t$ over the entire period is expressed as

$$\sum_{t \in [T-1]} \mathrm{tr}(C_t^\top P_t).$$

On the basis of the constraints and cost, the optimization problem to find the sales transition that minimizes the total cost incurred over the entire period is formulated as the following optimal transport problem:

$$\min_{\{P_t\}} \quad \sum_{t \in [T-1]} \mathrm{tr}(C_t^\top P_t) \tag{4a}$$
$$\text{s.t.} \quad P_t \geq O \qquad (t \in [T-1]), \tag{4b}$$
$$P_t \mathbf{1} = s_t \qquad (t \in [T-1]), \tag{4c}$$
$$\mathbf{1}^\top P_t = s_{t+1}^\top \qquad (t \in [T-1]). \tag{4d}$$

## Introducing regularization terms

In real markets, especially in matured or saturated markets, the tendencies of sales transitions between products do not change drastically from one period to the next [13,14]. In addition, some customers may exhibit uncertain behavior in purchasing products, influenced by various unobservable factors [15]. To take these behaviors into account, we incorporate two

regularization terms in Problem (4): one that promotes gradual changes in sales transition matrices across consecutive periods and another that allows for customers who do not switch products even when the cost is low.

**Period-related regularization.** Consumer behavior in real markets generally evolves gradually rather than changing abruptly, except in special cases [13,14]. To reflect this, we introduce a regularization term that encourages consecutive sales transition matrices to be similar:

$$\sum_{t\in[T-2]} \|P_{t+1} - P_t\|_F^2,$$

where $\|\cdot\|_F$ is the Frobenius norm defined as $\|P\|_F := \sqrt{\mathrm{tr}(P^\top P)}$ for a matrix $P$. We choose the Frobenius norm for its differentiability, but other norms may also be used. For instance, replacing it with the L1 norm yields the fused Lasso [16], which has been successfully applied in scenarios where adjacent variables are expected to change gradually for regression and classification tasks.

Several studies have highlighted the importance of temporal smoothness in estimating time-varying transition matrices in dynamic systems. Chiba et al. [8] employed a similar objective function so that the estimated transition matrices are close to each other for consecutive periods. Furthermore, time-series modeling literature [17–19] suggests that enforcing smooth transitions enhances model interpretability and performance, particularly in capturing gradual coefficient shifts. Empirical evidence from market dynamics research suggests that drastic changes in market shares are uncommon unless triggered by significant external shocks [5,20]. Thus, by imposing smoothness on transition matrices, we aim to consider the gradual evolution of purchasing behavior over time.

**Entropy regularization.** In Problem (4), we assume that when customers switch their purchasing products from one period to the next, they will select products with lower transition costs. However, in reality, purchasing behavior may involve some degree of stochastic uncertainty, influenced by various unobservable factors, and may not follow a strictly deterministic process. To model the inherent uncertainty in customer behavior, we introduce the following regularization term using the negative entropy function:

$$H(P_t) := -\sum_{n\in[N]}\sum_{m\in[N]} \frac{(P_t)_{nm}}{S_t}\left(1 - \log\frac{(P_t)_{nm}}{S_t}\right) \quad (t\in[T-1]), \tag{5}$$

where $S_t := \mathbf{1}^\top \boldsymbol{s}_t\ (t\in[T])$ is a constant representing the total sales in the period $t\in[T-1]$. By minimizing the negative entropy (5), we aim to avoid the concentration of sales transitions toward products with low transition costs.

From a market dynamics perspective, the regularization term (5) reflects the idea that brand-switching behavior is not determined solely by cost. Instead, some customers exhibit a certain degree of randomness in their choices, and the entropy has often been used to model this as a probabilistic model in brand-switching analysis [15,21,22]. Herniter [21] used entropy for analyzing brand-switching behavior, assuming customers' brand choices follow probabilistic distributions rather than deterministic rules, while Bass [15] emphasized the stochastic nature of consumer preferences. Kumar and Bector [22] further established the maximum entropy principle as a unifying framework for modeling brand-switching tendencies. Building on these existing studies, we incorporate the entropy regularization to capture the inherent randomness in consumer behavior, which cost-based models may not fully explain.

**Overall formulation.** The resulting optimization problem to estimate the sales transitions between products over time, incorporating the period-related and entropy regularizations, is formulated as follows:

$$\min_{\{P_t\}} \quad \sum_{t \in [T-1]} \mathrm{tr}(C_t^\top P_t) + \alpha \sum_{t \in [T-2]} \|P_{t+1} - P_t\|_F^2 + \beta \sum_{t \in [T-1]} H(P_t) \tag{6a}$$

$$\text{s.t.} \quad \text{Eqs (4b), (4c), and (4d),} \tag{6b}$$

where $\alpha, \beta \geq 0$ are the constants to control the strength of the regularization terms. Since the objective function of Problem (6) is a convex differentiable function, and the constraints are simple linear constraints. Thus, we use the projected gradient method to solve Problem (6).

## Projected gradient method

We describe the projected gradient method algorithm to solve Problem (6). For notational simplicity, we concatenate the decision variables $P_t$ ($t \in [T-1]$) into $\mathcal{P} := (P_t)_{t \in [T-1]} \in \mathbb{R}^{N \times N \times (T-1)}$. For Problem (6), we also represent its objective function as $f: \mathbb{R}^{N \times N \times (T-1)} \to \mathbb{R}$ and feasible region as $\mathcal{D} \subseteq \mathbb{R}^{N \times N \times (T-1)}$.

The projected gradient method is one of the optimization algorithms for convex optimization problems. In each iteration, it updates the solution by performing gradient descent and projection onto the feasible region. Let $\mathcal{P}^{(k)}$ denote the solution at the $k$-th iteration of the projected gradient method. In this case, the update rule of the projected gradient method is expressed as follows:

$$\mathcal{P}^{(k+1)} := \mathrm{Proj}_{\mathcal{D}}(\mathcal{P}^{(k)} - \eta_k \nabla f(\mathcal{P}^{(k)})), \tag{7}$$

where $\eta_k > 0$ represents the step size used in gradient descent and $\mathrm{Proj}_{\mathcal{D}}(\cdot)$ denotes the projection onto the feasible region $\mathcal{D}$. This update is repeated until the solutions $\mathcal{P}^{(k)}$ and $\mathcal{P}^{(k+1)}$ are sufficiently close, and finally, the final solution is output when the stopping condition is satisfied. We show the entire algorithm of the projected gradient method in Algorithm 1.

**Algorithm 1. Projected gradient method for problem (6).**

```
1: Set the initial point 𝒫⁽¹⁾.
2: Set k = 1.
3: while the stopping condition is not satisfied do
4:     Compute 𝒫′ = 𝒫⁽ᵏ⁾ − ηₖ∇f(𝒫⁽ᵏ⁾).
5:     Compute 𝒫⁽ᵏ⁺¹⁾ = Proj_𝒟 𝒫′.
6:     Set k := k + 1.
7: end while
8: Output 𝒫⁽ᵏ⁾.
```

Since the objective function $f$ is strictly convex and the feasible region $\mathcal{D}$ is a closed convex set, Problem (6) has the unique optimal solution $\mathcal{P}^* \in \mathcal{D}$. Following the analysis of the projected gradient method [23], we can bound the difference in objective values between the solutions $\{\mathcal{P}^{(k)}\}$ generated by Algorithm 1 and the optimal solution $\mathcal{P}^*$. For $\mathcal{P} = (P_t)_{t \in [T-1]} \in \mathbb{R}^{N \times N \times (T-1)}$, we define its norm as $\|\mathcal{P}\| := \sqrt{\sum_{t \in [T-1]} \|P_t\|_F^2}$.

**Theorem 1.** *Suppose* $\{\mathcal{P}^{(k)}\}$ *is the sequence of solutions generated by Algorithm 1. Let* $D :=$ $\|\mathcal{P}^{(0)} - \mathcal{P}^*\|$. *Then, for every positive integer* $k \geq 1$, *the following inequality holds:*

$$\min_{1 \leq s \leq k} f(\mathcal{P}^{(s)}) - f(\mathcal{P}^*) \leq \frac{D^2 + 2\sum_{s=1}^{k} \eta_s^2 \|\nabla f(\mathcal{P}^{(s)})\|^2}{\sum_{s=1}^{k} \eta_s}.$$

*Proof*: See the proof of Theorem 4.1 in Beck and Teboulle [23]. □

**Efficient computation.** In Algorithm 1, each iteration requires projecting $\mathcal{P}' := (P'_t)_{t \in [T-1]}$ onto the feasible region $\mathcal{D}$. This projection is performed by solving the following optimization problem:

$$\min_{\{P_t\}} \quad \sum_{t \in [T-1]} \|P'_t - P_t\|_F^2 \tag{8a}$$

$$\text{s.t.} \quad P_t \geq O \quad\quad (t \in [T-1]), \tag{8b}$$

$$P_t \mathbf{1} = \boldsymbol{s}_t \quad\quad (t \in [T-1]), \tag{8c}$$

$$\mathbf{1}^\top P_t = \boldsymbol{s}_{t+1}^\top \quad\quad (t \in [T-1]). \tag{8d}$$

Since the constraints for each period $t$ involve only $P_t$, Problem (8) decouples into $(T-1)$ subproblems—one per period. Each subproblem is convex quadratic optimization with linear constraints and can be solved independently by a powerful optimization solver (e.g., Gurobi or CPLEX). By solving each subproblem separately and then combining their solutions, we obtain $\mathcal{P}^{(k+1)}$, which is the projection of $\mathcal{P}'$ onto the feasible region $\mathcal{D}$. This decomposition reduces computation costs and allows efficient handling of large-sized problems.

## Numerical experiments

We conducted numerical experiments using POS data from Japanese supermarkets to verify the effectiveness of the proposed method. This dataset focuses on beverages and includes quarterly sales volumes, sales amounts, and product names from July 2013 to June 2022, enabling us to capture underlying puchase transitions under different market conditions. From this dataset, we set a cost matrix based on product prices, sales trends from the POS data, and the similarity of product names. To discuss the effectiveness of the proposed method, we examined the estimation results for two case studies: liquor tax changes and the launch of new coffee brands. We also conducted a sensitivity analysis of the cost matrix to evaluate the stability of the estimation results.

### Experimental setup

The POS data used in this experiment consists of beverage sales of alcoholic and coffee products collected from supermarkets across Japan from July 2013 to June 2022. This dataset does not include individual customer data, and we used quarterly sales volumes, sales amounts, and product names. For alcoholic beverages, we focused on the top eight sales categories (Happoshu, Beer, Shochu, Cocktails, Sake, Wine, Whisky/Brandy, and Liqueur). For coffee beverages, we looked at the top six brands by sales amount (Craft Boss, Georgia, Nescafe, Blendy, Wanda, and Fire). For the both alcoholic and coffee beverages, we aggregated the other products into a single category labeled "Others" to simplify the analysis. In addition, to account for customers who stopped purchasing or newly entered the market, we included a dummy category labeled "External" in both cases. For simplicity, we refer to alcoholic beverage categories and coffee brands collectively as *products* throughout this section.

## Cost matrix

Since the POS data does not include customer IDs, we constructed cost matrices $C_t$ ($t \in [T - 1]$) based on the available side information: product prices, sales trends, and the similarity of product names. In this experiment, we assume that the cost matrices are constant over time, meaning the same cost matrix is used for all $t \in [T - 1]$, i.e., $C_t = C$ ($t \in [T - 1]$). The construction of $C$ assumes that the cost of purchase transitions is low between products with similar names and prices, while the cost is high between products with similar sales trends. The details of how the cost matrix was constructed for this experiment are as follows:

Similarity of product names.  We used a pre-trained Japanese BERT model [24], which was trained on a Japanese corpus, to obtain vector representations for each product name, and then calculated the cosine similarity between the vectors of each product.

Similarity of sales trends.  For each product, we took the weekly differences in sales over 12 weeks to construct a vector representing the sales trend. We then calculated the cosine similarity of these vectors.

Similarity of prices.  We formed a three-dimensional vector containing its average price, the difference between the maximum and average prices, and the difference between the average and minimum prices for each product. We then calculated the cosine similarity of these vectors.

## Results

**Liquor tax changes in alcoholic beverages.** First, we examine how purchasing behavior changed after the liquor tax revision in July 2020, which reduced tax rates for Sake (Japanese rice wine) and Beer but increased rates for Happoshu (Happoshu is a type of low-malt alcoholic beverage classified separately from regular beer in Japan, benefiting from a lower tax rate than standard beer.) and Wine. Fig 2 presents the estimated purchase transitions among these alcoholic beverages from July–September to October–December 2020, visualized as a heatmap where each cell shows the ratio of transitions from one product to another. Overall, most cells on the diagonal are large and close to 1, indicating that the majority of purchases remained within the same product category.

Nevertheless, some off-diagonal entries reveal notable switching behaviors. To analyze these switches in more detail, we created a diagram (Fig 3) derived from the heatmap in Fig 2, focusing on the liquor categories whose tax rates changed in July 2020. Each node in Fig 3 represents a category, and the directed edges indicate the direction and proportion of purchase transitions. In Fig 3, a portion of customers of Happoshu switched to Sake and Beer, likely because the liquor tax revision increased the rate for Happoshu while lowering those for Sake and Beer, prompting consumers to choose a product with a lower tax burden. In contrast, even though the tax rate for Wine increased, we observed that some customers from other categories (e.g., External) migrated to Wine. One possible explanation is the annual release of Beaujolais Nouveau in November [25], which traditionally drives higher wine consumption in Japan [26]. Consequently, these results suggest that seasonal factors may encourage customers to purchase Wine, despite a tax increase.

**Breakthrough of craft boss in coffee beverages.** Craft Boss is a portable, PET-bottled coffee brand introduced in April 2017 [27]. It is known for its distinctive packaging, which is designed to attract consumers seeking a convenient alternative to traditional canned coffee. To investigate how purchasing behavior shifted after its launch, we examined the estimated purchase transitions for coffee products across three periods: July–September 2017, October–December 2017, and January–March 2018. We present the transitions from July–September to

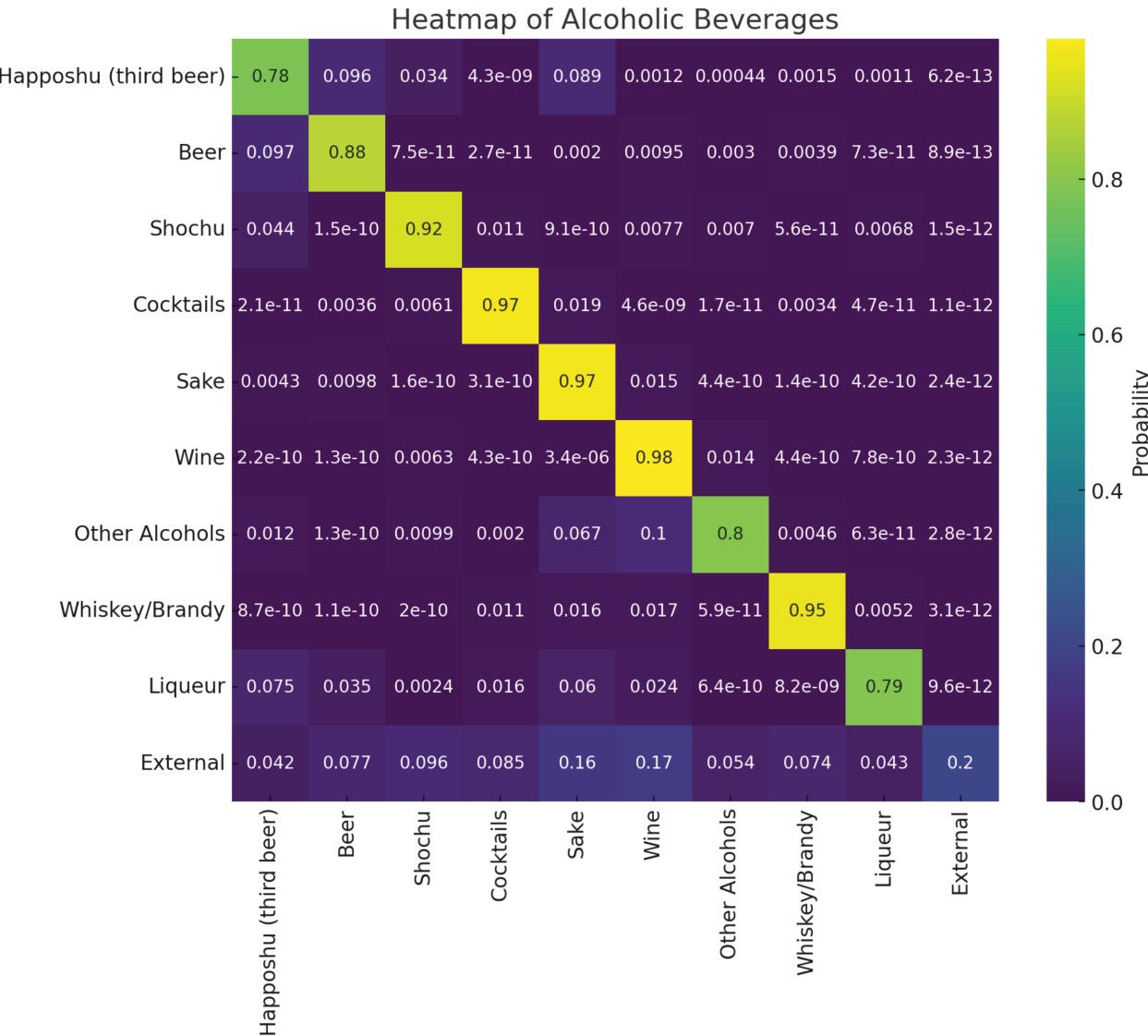

**Fig 2. Alcoholic product sales transition (July–September to October–December 2020).** The vertically and horizontally aligned axes represent the product categories for July–September and October–December 2020, respectively. Each cell indicates the ratio of sales transitions from row to column category. From this figure, we can see the transition from Happoshu to Sake and from External to Wine.

October–December 2017 in Fig 4 and from October–December 2017 to January–March 2018 in Fig 5, focusing on the top six coffee brands.

Figs 4 and 5 show notable shifts toward Craft Boss from other brands. In particular, the estimated transitions from Blendy and Nescafe to Craft Boss were relatively high, suggesting that the introduction of Craft Boss successfully attracted customers from established brands. Interestingly, while Blendy and Nescafe are also bottled coffee products, they are primarily designed for at-home consumption [28,29]. Thus, it seems that their target market differs from Craft Boss, which emphasizes portability. Nevertheless, our results indicate that Craft Boss attracted consumers from these brands, underscoring how a new product—especially

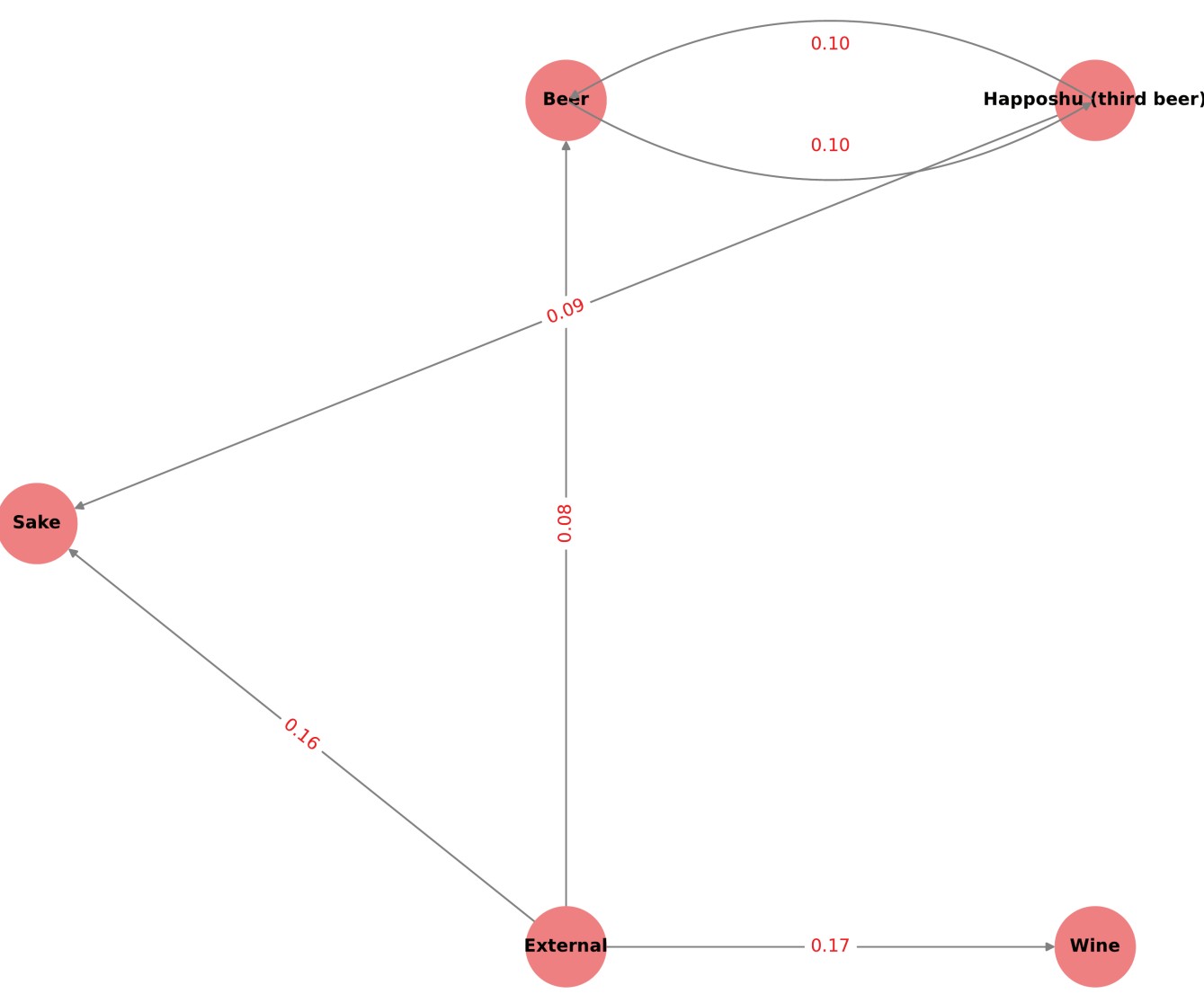

**Fig 3. Alcoholic sales transition (tax-related) (July–September to October–December 2020).** This figure illustrates the estimated purchase transitions among the categories impacted by the liquor tax revision from Fig 2. Each node represents the product categories purchased from July–September to October–December 2020. The directed edges indicate the direction and proportion of purchase transitions from the first to the second period.

one marketed around convenience—can disrupt existing segments and capture market share from the competitors.

**Sensitivity analysis and its results** To evaluate the stability of the estimation results, we conducted a sensitivity analysis by perturbing the cost matrix for the alcoholic beverages experiment. First, we constructed a perturbed cost matrix $\widetilde{C}_t$ for each $t \in [T-1]$ by adding a positive value $\delta > 0$ to each element of the original cost matrix $C_t$ from the previous experiment, which is shown in Fig 6. We then estimated sales transitions with $\widetilde{C}_t$ and compared the results with those obtained from the original cost matrix $C_t$. For this experiment, we set $\delta \in \{0.01, 0.1, 1\}$ to examine the effects of small to large perturbations on the estimation results.

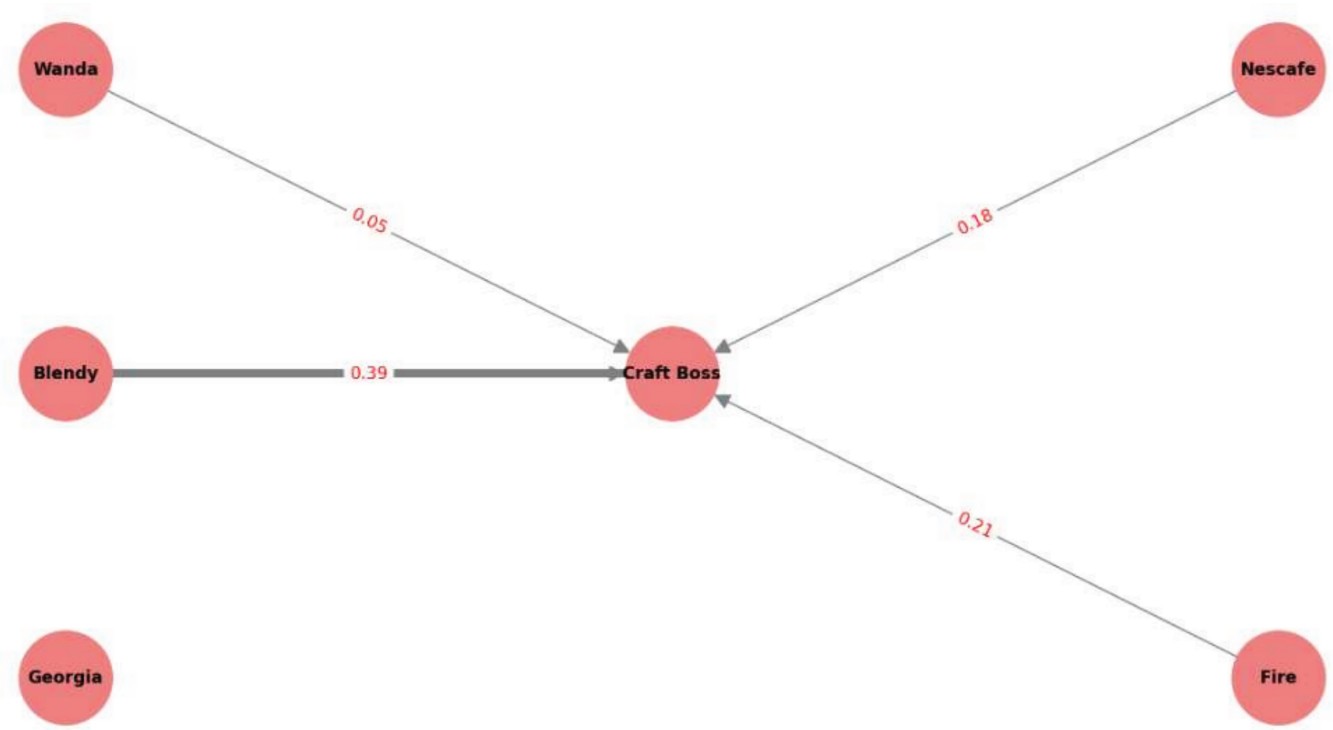

**Fig 4. Coffee product sales transition (July–September to October–December 2017).** This figure visualizes the estimated purchase transitions among the top six coffee brands from July–September to October–December 2017. From this figure, we can observe the notable transition from Blendy to Craft Boss.

Fig 7 shows the estimated sales transition for each perturbed cost matrix from August–October 2020 to November 2020–January 2021, focusing on the alcoholic beverages experiment. Fig 7 (i) displays the results based on the original cost matrix $C_t$, and the subsequent figures illustrate the results for the perturbed cost matrices $\widetilde{C}_t$ with $\delta \in \{0.01, 0.1, 1\}$. In addition, Table 1 shows the Frobenius norms of the differences between the sales transition matrices estimated from the original cost $C_t$ and those estimated from the perturbed cost $\widetilde{C}_t$ for each $\delta$. From Fig 7, we observe the tendency that as $\delta$ increases, the estimation results deviate more from the original ones, which can be also confirmed by the results in Table 1. Figs 7(ii) to 7(iv) indicate that for $\delta = 0.01$ and $0.1$, the estimated transitions exhibit similar tendencies to the original results, whereas a notable change occurs for $\delta = 1$. These results suggest that our method remains stable under small perturbations but can be sensitive to larger perturbations.

## Discussion

We discuss the effectiveness and implications of the proposed method from our experimental results.

**Implications for marketing actions and policy decisions.** Our results show that, following the liquor tax revision, many customers switched to products with lower taxes. This finding points to the importance of flexible pricing strategies and discounts, especially in product categories strongly affected by tax changes. Adjusting prices quickly can help companies stay competitive. In addition, the significant impact of Craft Boss's market entry suggests that convenient packaging is a key factor in attracting customers from established brands.

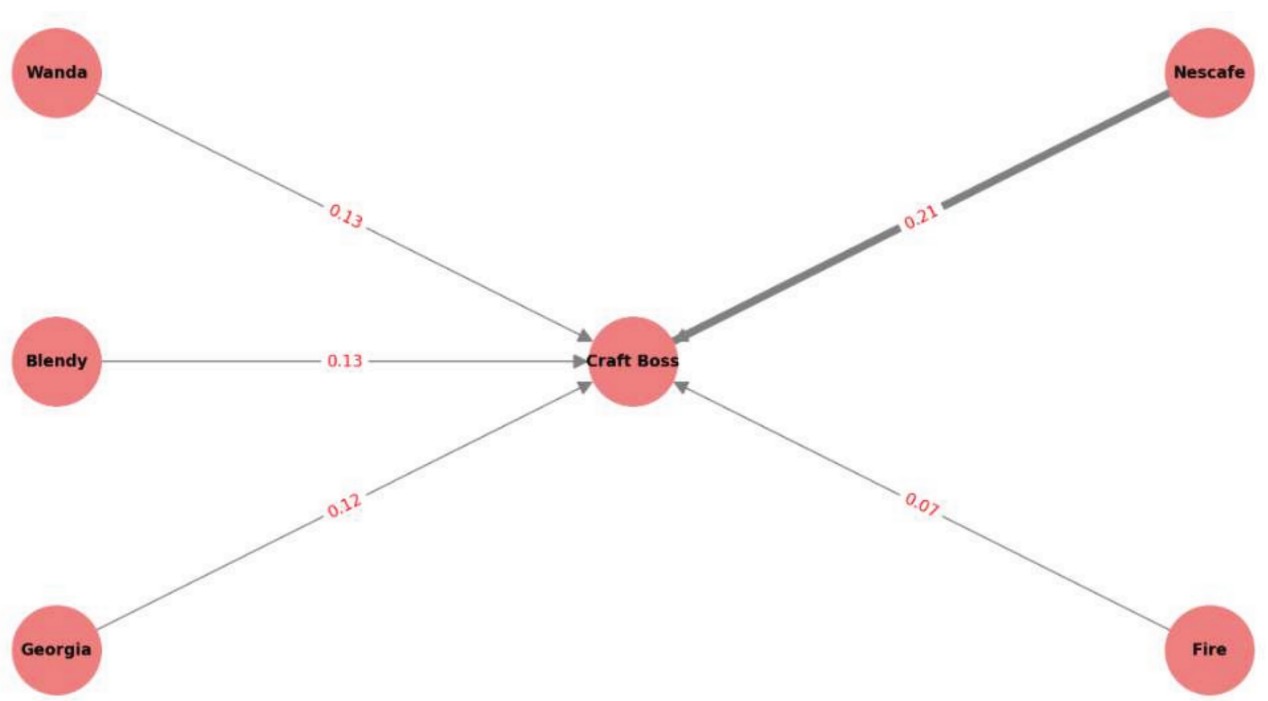

**Fig 5. Coffee product sales transition (October–December 2017 to January–March 2018).** This figure visualizes the estimated purchase transitions among the top six coffee brands from October–December 2017 to January–March 2018. Contrary to Fig 4, the transition from Nescafe to Craft Boss was remarkable.

Although our study looked at the Japanese beverage industry, many mature markets face similar challenges—such as strict privacy rules, limited overall growth, and tough competition among major players. The way we measure changes in consumer behavior, including the effects of new product launches or policy revisions, could be applied in such settings. By analyzing the impact of liquor tax changes on sales, our approach can also help policymakers design more effective policies and allow companies to plan better for shifts in consumer demand.

**Limitations of our evaluation.** While our experiments demonstrated the effectiveness of the proposed method, we recognize certain limitations that may affect its applicability. First, our method relies on a predefined cost matrix. If constructing reasonable cost matrices is infeasible or difficult due to insufficient data, our method may not be applicable, or its estimation validity may be compromised. In our experiment, we constructed the cost matrix based on available side information, such as product prices, sales trends, and product names. While we qualitatively examined the estimated results, further evaluation is needed to confirm the validity and reliability of the cost matrix we used, particularly through additional data sources.

Second, our proposed model (6) includes a period-related regularization and an entropy term in the objective function. However, their effectiveness in improving the accuracy of estimated purchase transitions has not been quantitatively evaluated. Since the POS data used in this study lacks customer IDs, we were unable to directly verify whether the estimated transitions reflect actual purchasing behavior. While we qualitatively assessed the validity of the results, we could not confirm whether the regularization terms genuinely enhance estimation

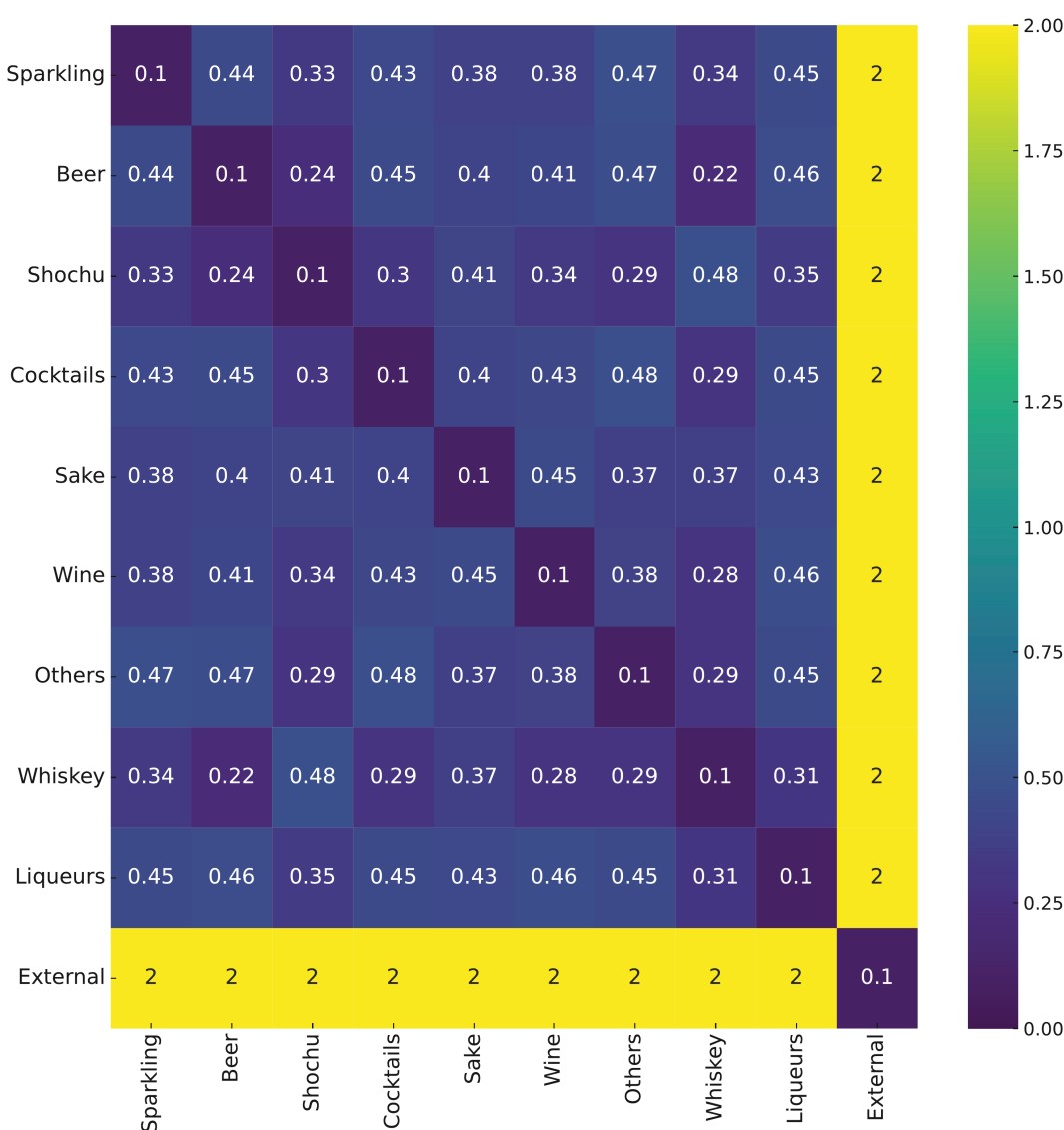

**Fig 6. Cost matrix used in the sensitivity analysis.** Vertically and horizontally aligned cells represent the product categories, and each cell shows the cost of transitioning from row to column products.

accuracy. To rigorously evaluate their impact, future work should use POS data with customer IDs to examine how these terms influence the estimation results.

Finally, our proposed model (6) adopts a macroscopic perspective and does not account for individual differences among customers. Specifically, we assume that the cost matrix is shared among all customers without considering personal factors such as brand loyalty and individual preferences. In reality, purchasing behavior is influenced by various factors, including customer preferences, brand loyalty, and advertising, leading to brand-switching behaviors that depend on customer-specific costs. Future research should explore methods to capture customer-specific transition costs from aggregated POS data, such as clustering customers based on purchasing patterns or leveraging external demographic information, even when customer IDs are unavailable.

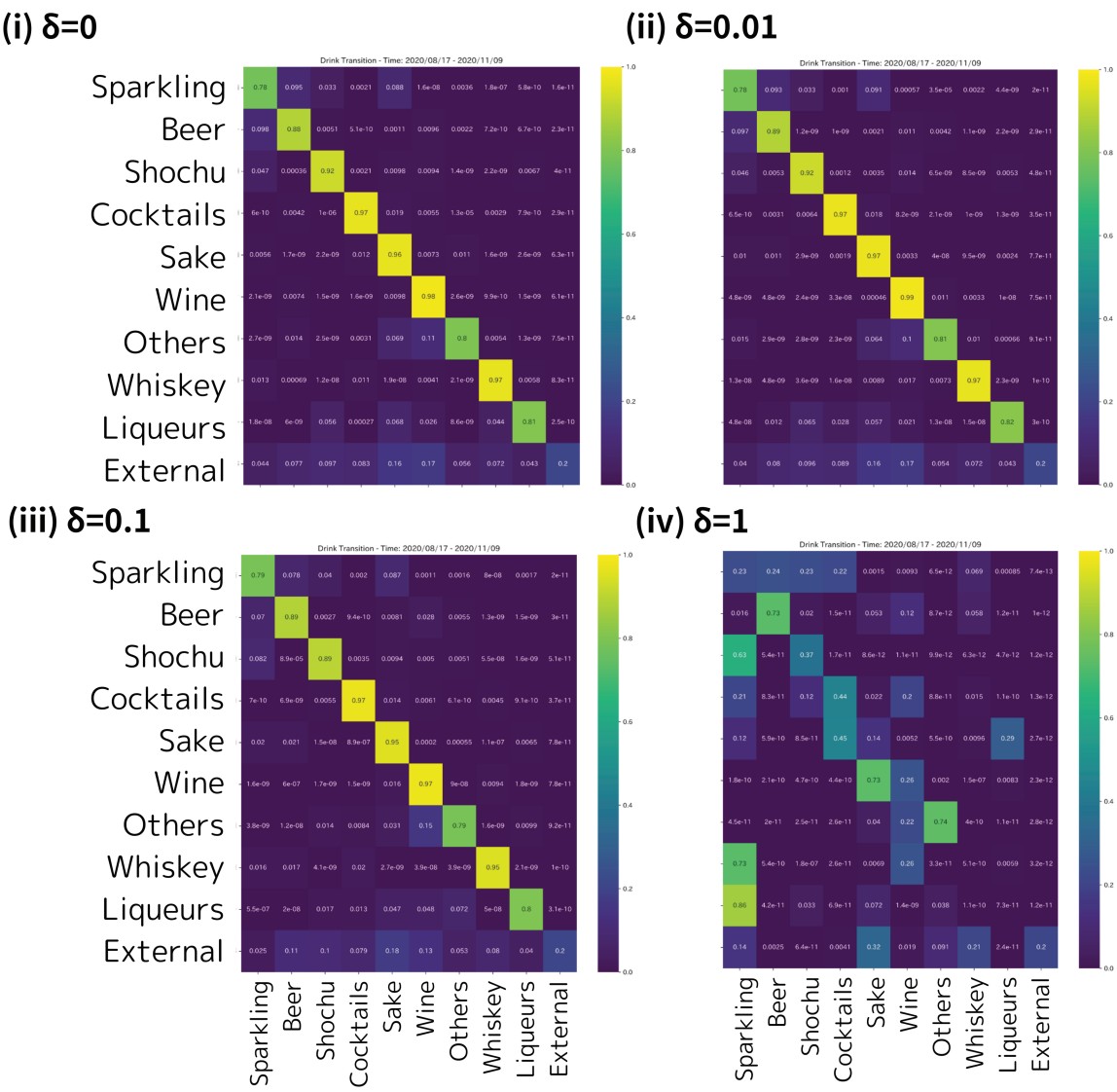

**Fig 7. Alcoholic product sales transition with perturbed cost matrices (August–October 2020 to November 2020–January 2021).**
The estimated purchase transitions among alcoholic products from August to November 2020 are visualized for the original cost matrix and perturbed cost matrices with $\delta = 0.01, 0.1,$ and 1. This figure shows that for $\delta = 0.001, 0.1$, the estimation results have similar tendencies to the original results, whereas a notable change occurs for $\delta = 1$.

**Table 1. Frobenius norm of the differences in sales transition matrices from the original estimation.**

|  | $\delta = 0.01$ | $\delta = 0.1$ | $\delta = 1$ |
|---|---|---|---|
| Frobenius Norm | 0.0728 | 0.1517 | 2.5565 |

## Conclusion

In this study, we introduced a method to estimate purchase transitions between products without requiring customer identifiers. Our approach models the cost of switching between

products as an optimal transport problem, using aggregated sales data and two regularization terms: one that encourages similarity in purchase transitions across consecutive periods, and another that accounts for customers who may not behave strictly according to cost minimization. In addition, we used a projected gradient method to solve this problem.

Applying our method to Japanese beverage POS data revealed its capability to capture key market shifts, notably customers' movement to lower-tax products following a liquor tax revision and a marked shift toward a newly launched coffee brand. Sensitivity analysis further showed stable estimation results against small perturbations in the cost matrix. These findings suggest that the proposed method can effectively illuminate competitive relationships and market dynamics, even when customer-level data is unavailable. Given heightened privacy concerns in many industries, our approach could help companies maintain competitiveness in mature markets by revealing patterns in product-switching behavior at an aggregate level.

Despite these promising outcomes, several future directions remain. First, we need to explore more accurate methods for constructing reasonable cost matrices, potentially by leveraging partial customer purchase transitions or conducting customer surveys. Second, a more thorough evaluation of the role of our regularization terms—potentially using datasets with customer identifiers—would help validate the model's accuracy. Finally, incorporating richer attributes such as brand loyalty and personal preferences could enhance the model's ability to capture individual-level behavior. Pursuing these avenues will further increase the robustness and applicability of our method, strengthening its value as a privacy-preserving tool for analyzing market structure and consumer trends.

## Supporting information

**S1 Appendix. Hyperparameters used in the experiment.**
(PDF)

## Acknowledgments

This study was conducted as a part of the Data Analysis Competition hosted by the Joint Association Study Group of Management Science. The authors would like to thank the organizers and Nikkei Inc. for providing us with a real data set.

## Author contributions

**Conceptualization:** Shoki Yamao, Ryota Ueda, Shoichiro Koguchi, Michi Nakase, Aru Suzuki, Kohdai Toyoda, Ken Kobayashi, Kazuhide Nakata.

**Data curation:** Shoki Yamao, Ryota Ueda, Shoichiro Koguchi, Michi Nakase, Aru Suzuki, Kohdai Toyoda, Ken Kobayashi, Kazuhide Nakata.

**Formal analysis:** Shoki Yamao, Ryota Ueda, Shoichiro Koguchi, Michi Nakase, Aru Suzuki, Kohdai Toyoda, Ken Kobayashi, Kazuhide Nakata.

**Investigation:** Shoki Yamao, Ryota Ueda, Shoichiro Koguchi, Michi Nakase, Aru Suzuki, Kohdai Toyoda, Ken Kobayashi, Kazuhide Nakata.

**Methodology:** Shoki Yamao, Ryota Ueda, Shoichiro Koguchi, Michi Nakase, Aru Suzuki, Kohdai Toyoda, Ken Kobayashi, Kazuhide Nakata.

**Project administration:** Shoki Yamao.

**Supervision:** Ken Kobayashi, Kazuhide Nakata.

**Writing – original draft:** Shoki Yamao, Ryota Ueda, Shoichiro Koguchi, Michi Nakase, Aru Suzuki, Kohdai Toyoda, Ken Kobayashi.

**Writing – review & editing:** Shoki Yamao, Ryota Ueda, Shoichiro Koguchi, Michi Nakase, Aru Suzuki, Kohdai Toyoda, Ken Kobayashi, Kazuhide Nakata.

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
