## [Decision Letter · Decision Letter 0]

17 Dec 2024

PONE-D-24-49715Estimating Sales Transitions between Competing Products via Optimal TransportPLOS ONE

Dear Dr. Yamao,

Thank you for submitting your manuscript to PLOS ONE. After careful consideration, we feel that it has merit but does not fully meet PLOS ONE’s publication criteria as it currently stands. Therefore, we invite you to submit a revised version of the manuscript that addresses the points raised during the review process.

We look forward to receiving your revised manuscript.

Kind regards,

Takayuki Mizuno, Ph. D.

Academic Editor

PLOS ONE

Journal Requirements:

When submitting your revision, we need you to address these additional requirements. 1. Please ensure that your manuscript meets PLOS ONE's style requirements, including those for file naming. The PLOS ONE style templates can be found at https://journals.plos.org/plosone/s/file?id=wjVg/PLOSOne_formatting_sample_main_body.pdf and https://journals.plos.org/plosone/s/file?id=ba62/PLOSOne_formatting_sample_title_authors_affiliations.pdf 2. PLOS requires an ORCID iD for the corresponding author in Editorial Manager on papers submitted after December 6th, 2016. Please ensure that you have an ORCID iD and that it is validated in Editorial Manager. To do this, go to ‘Update my Information’ (in the upper left-hand corner of the main menu), and click on the Fetch/Validate link next to the ORCID field. This will take you to the ORCID site and allow you to create a new iD or authenticate a pre-existing iD in Editorial Manager. 3. We note that you have indicated that there are restrictions to data sharing for this study. PLOS only allows data to be available upon request if there are legal or ethical restrictions on sharing data publicly. For more information on unacceptable data access restrictions, please see http://journals.plos.org/plosone/s/data-availability#loc-unacceptable-data-access-restrictions.  Before we proceed with your manuscript, please address the following prompts: a) If there are ethical or legal restrictions on sharing a de-identified data set, please explain them in detail (e.g., data contain potentially identifying or sensitive patient information, data are owned by a third-party organization, etc.) and who has imposed them (e.g., a Research Ethics Committee or Institutional Review Board, etc.). Please also provide contact information for a data access committee, ethics committee, or other institutional body to which data requests may be sent. b) If there are no restrictions, please upload the minimal anonymized data set necessary to replicate your study findings to a stable, public repository and provide us with the relevant URLs, DOIs, or accession numbers. For a list of recommended repositories, please seehttps://journals.plos.org/plosone/s/recommended-repositories. You also have the option of uploading the data as Supporting Information files, but we would recommend depositing data directly to a data repository if possible. We will update your Data Availability statement on your behalf to reflect the information you provide.

Reviewers' comments:

Reviewer's Responses to Questions

**Comments to the Author**

1. Is the manuscript technically sound, and do the data support the conclusions?

Reviewer #1: Yes

Reviewer #2: Partly

2. Has the statistical analysis been performed appropriately and rigorously? 

Reviewer #1: Yes

Reviewer #2: I Don't Know

3. Have the authors made all data underlying the findings in their manuscript fully available?

Reviewer #1: No

Reviewer #2: No

4. Is the manuscript presented in an intelligible fashion and written in standard English?

Reviewer #1: Yes

Reviewer #2: Yes

5. Review Comments to the Author

Reviewer #1: 1. The abstract could briefly mention the limitations of the study, such as the proprietary nature of the data or potential improvements using customer identifiers.

2. I would suggest including a flowchart or diagram summarizing the methodology for easier comprehension.

3. Heatmaps and transition diagrams are valuable but lack detailed captions. I would suggest that you explain what each axis represents and highlight the key observations in each figure.

Reviewer #2: General comment –First, I appreciate the opportunity to review this manuscript, which explores the estimation of sales transitions between competing products using an optimal transport framework. This is a timely and relevant study, particularly in mature markets where understanding customer switching behaviour is critical for companies seeking to maintain market share under increasingly stringent privacy regulations.

The manuscript presents an innovative approach by framing sales transitions as an optimal transport problem, complemented by thoughtful regularisations to reflect realistic market dynamics. The integration of a projected gradient method to solve the optimisation problem and its validation with real-world POS data from the Japanese beverage industry adds significant practical value to the study.

However, while the manuscript offers a promising contribution, certain areas require improvement to enhance its clarity, robustness, and overall impact. The theoretical rationale for framing the problem as an optimal transport task could be articulated more clearly, as this would help establish a stronger foundation for the proposed methodology. Additionally, further justification of the two regularisation assumptions and their implications on market behaviour would strengthen the argument.

In the methodology section, greater detail regarding the implementation of the projected gradient method, including computational considerations and convergence properties, would enhance the transparency and reproducibility of the results. Furthermore, while the case studies provide valuable insights, deeper discussion on the generalisability of the findings beyond the Japanese beverage industry would improve the study’s relevance to broader markets.

I hope the authors will find these comments constructive and useful in refining their manuscript for publication.

Comment 1 – The introduction provides a clear context for the study and highlights the importance of understanding customer transitions in mature markets. However, the articulation of research gaps could be more robust. While the authors note the increasing difficulty of obtaining purchase history data with customer identifiers due to privacy regulations, this is not sufficiently linked to broader gaps in the existing literature. For instance, the discussion on previous studies (lines 22–42) largely summarises methodologies without critically addressing their limitations in terms of applicability, scalability, or assumptions in real-world contexts. Moreover, while the authors mention challenges with the trivial solution in Chiba et al.'s linear programming method, there is insufficient discussion on why these limitations necessitate the proposed optimal transport framework and how it uniquely addresses unresolved questions in the field. Strengthening the argument for the novelty and necessity of this study would enhance the introduction significantly.

Comment 2 – The manuscript lacks a dedicated literature review or theoretical foundation section between the introduction and methodology. While the introduction briefly summarises prior work on brand-switching and related methods (lines 22–42), this does not sufficiently review the broader body of literature or establish the theoretical underpinnings of the study. A standalone section could elaborate on key concepts such as optimal transport theory, brand-switching behaviour, and market dynamics, providing deeper insight into how these areas interconnect. Additionally, a comprehensive discussion of relevant studies, including their contributions and limitations, would better contextualise the research gap and justify the novelty of the proposed method. This would strengthen the foundation for the study and enhance its academic contribution.

Comment 3 – The manuscript does not include a dedicated discussion section, as it transitions directly from numerical experiment results to the conclusion. This omission limits the opportunity to critically interpret the findings, connect them to the broader literature, and explore their implications. For example, the results indicating shifts in market share due to liquor tax reforms and the launch of "Craft Boss" could be more deeply analysed to highlight their relevance to market strategy and policy-making. Furthermore, the discussion could address the robustness and potential limitations of the proposed method, such as the reliance on the cost matrix construction and assumptions about purchasing behaviour. Including a discussion section would allow the authors to better contextualise their contributions, acknowledge the study's limitations, and suggest practical applications and future research directions. This would significantly enhance the manuscript's clarity and impact.

Comment 4 – The conclusion section does not conform to academic standards as it predominantly reiterates details from the methodology and results without providing a concise and impactful summary. A well-structured academic conclusion should:

1. Summarise the Study in a Concise Manner – Provide a clear and succinct recap of the research objectives, methods, and key findings, avoiding technical details that belong in earlier sections.

2. Highlight the Significance of the Study – Emphasise the overarching contribution of the research to the field without introducing new information or results.

3. Provide Closure – Offer a final, thoughtful remark that underscores the importance of the findings in a broader context, leaving the reader with a sense of completion.

Currently, the conclusion repeats details from the methodology and results rather than summarising the study holistically. To meet academic expectations, the authors should reframe this section to focus on the broader implications of their findings and provide a concise yet impactful closure.

6. PLOS authors have the option to publish the peer review history of their article (what does this mean?). If published, this will include your full peer review and any attached files.

Reviewer #1: **Yes: **Dr. Morakinyo Dada

Reviewer #2: No

---

## [Decision Letter · Decision Letter 1]

9 May 2025

Estimating Sales Transitions between Competing Products via Optimal Transport

PONE-D-24-49715R1

Dear Dr. Yamao,

We’re pleased to inform you that your manuscript has been judged scientifically suitable for publication and will be formally accepted for publication once it meets all outstanding technical requirements.

Kind regards,

Takayuki Mizuno, Ph. D.

Academic Editor

PLOS ONE

Additional Editor Comments (optional):

Reviewers' comments:

Reviewer's Responses to Questions

**Comments to the Author**

1. If the authors have adequately addressed your comments raised in a previous round of review and you feel that this manuscript is now acceptable for publication, you may indicate that here to bypass the “Comments to the Author” section, enter your conflict of interest statement in the “Confidential to Editor” section, and submit your "Accept" recommendation.

Reviewer #1: All comments have been addressed

Reviewer #3: All comments have been addressed

2. Is the manuscript technically sound, and do the data support the conclusions?

Reviewer #1: Yes

Reviewer #3: Yes

3. Has the statistical analysis been performed appropriately and rigorously? 

Reviewer #1: Yes

Reviewer #3: Yes

4. Have the authors made all data underlying the findings in their manuscript fully available?

Reviewer #1: Yes

Reviewer #3: Yes

5. Is the manuscript presented in an intelligible fashion and written in standard English?

Reviewer #1: Yes

Reviewer #3: Yes

6. Review Comments to the Author

Reviewer #1: The amendment made to the document is appropriate and addresses the concerns previously highlighted. It reflects a clear understanding of the feedback provided and demonstrates a thoughtful approach to refining the content in alignment with the objectives. The revised sections shows improved clarity, coherence, and relevance, ensuring that it aligns more effectively with academic expectations and industry standards. Additionally, the language used is more precise and the structure now facilitates better comprehension. Overall, the amendment significantly contributes to strengthening the paper's academic integrity. I am satisfied that the revised version is now suitably aligned with the intended purpose and meets the requirements of the review process. No further major revisions are deemed necessary at this point.

Reviewer #3: (No Response)

7. PLOS authors have the option to publish the peer review history of their article (what does this mean?). If published, this will include your full peer review and any attached files.

Reviewer #1: No

Reviewer #3: No

---

## [Editor Report · Acceptance letter]

PONE-D-24-49715R1

PLOS ONE

Dear Dr. Yamao,

I'm pleased to inform you that your manuscript has been deemed suitable for publication in PLOS ONE. Congratulations! Your manuscript is now being handed over to our production team.

Kind regards,

on behalf of

Dr. Takayuki Mizuno

Academic Editor

PLOS ONE